# Genome-Wide Analysis Reveals Transcription Factors Regulated by Spider-Mite Feeding in Cucumber (*Cucumis*
*sativus*)

**DOI:** 10.3390/plants9081014

**Published:** 2020-08-11

**Authors:** Jun He, Harro J. Bouwmeester, Marcel Dicke, Iris F. Kappers

**Affiliations:** 1Citrus Research Institute, Southwest University, Chongqing 400712, China; 2Laboratory of Plant Physiology, Wageningen University and Research, 6708 PB Wageningen, The Netherlands; H.J.Bouwmeester@uva.nl; 3Plant Hormone Biology Group, Swammerdam Institute for Life Sciences, University of Amsterdam, 1000 BE Amsterdam, The Netherlands; 4Laboratory of Entomology, Wageningen University and Research, 6708 PB Wageningen, The Netherlands; marcel.dicke@wur.nl

**Keywords:** cucumber, spider mite, transcription factor, promoter, *cis*-acting regulatory elements

## Abstract

To gain insight into the regulatory networks that underlie the induced defense in cucumber against spider mites, genes encoding transcription factors (TFs) were identified in the cucumber (*Cucumis*
*sativus*) genome and their regulation by two-spotted spider mite (*Tetranychus*
*urticae*) herbivory was analyzed using RNA-seq. Of the total 1212 annotated TF genes in the cucumber genome, 119 were differentially regulated upon spider-mite herbivory during a period of 3 days. These TF genes belong to different categories but the *MYB*, *bHLH*, *AP2/ERF* and *WRKY* families had the highest relative numbers of differentially expressed genes. Correlation analysis of the expression of TF genes with defense-associated genes during herbivory and pathogen infestation, and in different organs resulted in the putative identification of regulators of herbivore-induced terpenoid and green-leaf-volatile biosynthesis. Analysis of the *cis*-acting regulatory elements (CAREs) present in the promoter regions of the genes responsive to spider-mite feeding revealed potential TF regulators. This study describes the TF genes in cucumber that are potentially involved in the regulation of induced defense against herbivory by spider mites.

## 1. Introduction

Transcription factors (TFs) are DNA-binding proteins that recognize and bind to specific regulatory sequences, the *cis-*acting regulatory elements (CAREs, also called CREs for *cis*-regulatory elements), in the promoter of the target genes [1,2,3]. CAREs are typically located adjacent to the promoter region of a gene, but can also be found in the gene itself, in introns or even further away from the gene, upstream or downstream from the promoter [2]. By binding to these elements, TFs can stabilize or block the binding of RNA polymerase to DNA [4], catalyze the acetylation or deacetylation of histone proteins [5] or recruit co-activator or co-repressor proteins to form TF DNA complexes [6]. All these processes can result in a change in the expression of the target gene of the corresponding TF.

TFs have been studied in the available plant genome sequences. For example, about 8–9% of the 27,411 protein-encoding genes in the Arabidopsis genome (TAIR10) were identified as putative TFs [7,8,9]. In the rice genome, about 10% of the predicted 22,896 genes (MSU Rice Genome Annotation, Release 7) were suggested to encode TFs and in tomato, about 6% of the genes in the genome putatively encode TFs [10]. In cucumber, 1575 putative TFs were identified in the genome in the iTAK database [8], which represents about 7% of the 23,248 predicted genes ([11,12]; version 2.0).

TFs play a role in the response of plants to various environmental stresses including the attack by herbivores. Plants perceive this attack via damage- or herbivore-associated molecular patterns (DAMPS or HAMPS, [13]). Upon recognition of the attack, plant hormones moderate the response to the different attackers [14,15]. Subsequently, phytohormone signaling is translated into the activation or repression of gene expression. This transcriptional regulation usually depends on the transcription and translation of TFs [16], although other factors such as chromatin-remodeling could also be involved [17]. The transcriptional changes in, for example, metabolism-associated genes finally result in an altered metabolic profile of the attacked plant [16,18].

Transcription factors can be classified into different families according to the conserved DNA-binding domains and over sixty families of TFs have been identified in various plant species [3,19]. Among them, particularly the MYB, bHLH, WRKY, AP2/ERF, NAC and bZIP families have been shown to play a role in plant defense [20]. For example, MYBs are involved in the regulation of the expression of genes involved in metabolism and inducible defenses in Arabidopsis [21,22,23], *Populus tremuloides* [24] and *Nicotiana attenuata* [25]. bHLH TFs in Arabidopsis function in regulation of jasmonic acid (JA) signaling pathway [26,27,28], flavonoid biosynthesis [29] and defense to herbivores such as *Helicoverpa armigera* [30] and *Spodoptera littoralis* [31]. Many ERF TFs are essential regulators of plant responses to biotic and abiotic stress. Arabidopsis AtERF5, AtERF6 and AtRAP2.2 have been implicated in resistance to *Botrytis cinerea* mediated by JA and ethylene [32,33]. In rice, OsERF3 is essential for resistance to the chewing striped stem borer *Chilo suppressalis* via transcriptional regulation of two MAPKs and two WRKY TFs, and mediation of JA and salicylic acid signaling [34]. WRKY TFs are of major importance in plant responses to various biotic actors including bacteria [35], fungi [36] and herbivores [37,38].

Here, we studied the involvement of TFs and CAREs in cucumber in the response to the two-spotted spider mite (*Tetranychus urticae*). Cucumber is an important vegetable grown and consumed world-wide (http://faostat.fao.org). The chelicerate two-spotted spider mite is a major pest that feeds on plants in more than 140 plant families [39] including important agricultural crops such as maize, grape, tomato, pepper and cucumber. Global transcriptional changes in response to feeding by spider mites were demonstrated to occur in plant species such as Arabidopsis [40], tomato [41], grapevine [42] and cassava [43] with a number of differentially expressed genes (DEGs), which included hundreds of TF genes. Previously, we have analyzed the transcriptional changes in cucumber during the first three days of spider-mite feeding and identified more than 2000 DEGs [44]. Here, we focus on the regulation of the expression of TF genes by spider-mite herbivory and include a comparison with the response upon infection with a pathogen. Furthermore, we analyze promoter sequences of genes that are affected in their expression due to feeding by spider mites for possible TF binding sites. 

## 2. Results

### 2.1. TF Genes in the Cucumber Genome

Based on the annotated set of genes predicted from the cucumber genome (http://www.icugi.org/cgi-bin/ICuGI/index.cgi), 1212 genes with a putative TF function of different classes were retrieved (Figure 1). Half of all TF genes belong to only four different TF gene families, i.e., MYB (189), bHLH (149), AP2/ERF (143) and C2H2 zinc fingers (131). Furthermore, among the 1212 TF genes there are 84 genes encoding NAC domain proteins, 73 genes encoding bZIP (basic-leucine zipper domain) proteins and 67 genes encoding WRKYs. In addition, there are 10 TF gene families with less than 50 members each, including MADS (41 genes), GRAS (37 genes), WTHT (33 genes), TCP (27 genes), GATA (26 genes), NF-Y (26 genes), ZF-TF (22 genes), ARF (20 genes), HOX (17 genes) and 99 putative TFs that do not belong to any of the above TF families.

### 2.2. Cucumber Genes Responsive to Spider-Mite Feeding

Previously, using an RNA-seq approach, we identified 2348 genes that were differentially expressed between spider mite-infested and non-infested plants of two genotypes of cucumber, accession Chinese long with bitter and accession Corona with non-bitter foliage [44]. All DEGs were organized into four groups based on their expression at different time points after feeding (Figure 2). In group I, gene expression increased during the period of feeding in both accessions whereas group IV contained genes for which expression decreased in both accessions. Groups II and III include those genes for which transcripts remained more or less unaltered in Chinese long but increased (group II) or decreased (group III) in Corona after spider-mite feeding. Statistical analysis showed that both duration of feeding and the genotype had significant effects on the transcript abundance of genes and there was a significant interaction between both factors in three of the four groups of genes (Figure 2). We used this collection of DEGs as a data set to identify TFs that play a role in both positive and negative early responses towards two-spotted spider mites in cucumber. We compared our results with the transcriptional response of cucumber towards the pathogen *Pseudoperonospora cubensis* downy mildew as described by Adhikari, et al. [45] to study similarities and differences in the involvement of TFs in the response of cucumber to these two different biotic stresses ([44], Table 1).

### 2.3. Regulation of TF Expression in Response to Spider-Mite Herbivory

Out of the 1212 TF-encoding genes (Figure 1), transcription of 119 was differentially regulated by spider-mite feeding, of which 60 were generally upregulated (in group I and group II DEGs) and 59 were downregulated (in groups III and IV). The distribution of the number of genes in the different groups of DEGs over the TF gene families was analyzed (Table 1) to discover general trends in the expression of these genes. Groups I and II, both representing increased gene transcript levels after spider-mite feeding, included five MYBs. In contrast, seven MYBs were present in group III and two in group IV, which both represent genes for which transcription is repressed by spider-mite feeding. TF genes of the bHLH family were almost equally distributed over the four groups, with 4 to 6 genes in each of the groups. Most of the ERF TF genes were included in group I (4) and group II (13) and displayed increased transcription upon spider-mite feeding, while seven ERF genes displayed decreasing transcription, six in group III and one in group IV. Most of the WRKY genes were assigned to group III or group IV, and thus displayed decreasing transcription in response to spider-mite feeding. Taken together it seems that ERFs and WRKYs were preferentially up- and downregulated by spider-mite herbivory, respectively, while MYBs and bHLHs were about equally up- and downregulated. The other TF families were basically randomly distributed over the four groups of DEGs (Table 1).

As the WRKY, AP2/ERF, bHLH and MYB TF genes were most responsive to spider mites and were also reported to encode important TFs in plant defense [20], we made a more detailed analysis of their expression pattern during early spider-mite feeding (Figure 3). Expression of MYBs and bHLHs during the first three days after the onset of spider-mite feeding was equally up- and downregulated (Figure 3), obviously corresponding to their classification in the DEG self-organizing maps (Figure 1). Most of the AP2/ERFs were found to be upregulated after spider-mite feeding, particularly in Corona. The majority of WRKYs were found in group III and thus were suppressed by spider-mite feeding. However, Figure 3 shows that expression of most of the WRKYs actually increased on the first or second day of spider-mite feeding before transcripts were downregulated on the third day.

### 2.4. TF Genes Most Strongly Influenced by Spider-Mite Feeding

Spider-mite feeding resulted in differential expression of TF genes in both directions. The relative transcript abundance of a putative bHLH gene (Csa3M002860) increased about 20-fold (log_2_Ration ≈ 4.4) one day after the onset of spider-mite feeding compared to transcription in non-infested leaves making it the most strongly affected TF gene in spider mite-infested cucumber leaves. Although the absolute abundance of the transcripts of this TF gene decreased on days 2 and 3, expression was still about 14- and 10-fold higher than in non-infested plants, respectively (Figure 4). A similar pattern was found for a MYB TF gene, Csa6M121970, with the strongest induction (18-fold) on day 1, and about 7- and 6-fold higher compared with non-infested plants on succeeding days. The other putative TF genes that are depicted in the right panel of Figure 4 have the strongest induction of expression on different days, but all belong to the ten most upregulated TFs. This group includes three ERFs. In contrast, the two strongest downregulated TF genes belong to the bHLH family (Figure 4, left panel). Transcription of the bHLH TF gene, Csa3M850530, first increased slightly after one day of feeding and subsequently decreased to approximately half the amount of transcripts of non-infested leaves on day 2 and to 8% (log_2_Ration ≈ 3.6) on day 3. Expression of the bHLH TF gene, Csa7M452040, showed a similar pattern, resulting in a decrease in transcripts to 10% (log_2_Ratio ≈ 3.3) compared to non-infested plants after three days. The expression of the ten strongest suppressed TF genes decreased mainly on day 3 and this category includes three MYBs.

### 2.5. Co-Expression of TF Genes with Defense-Related Genes

Co-expression analysis between TF genes and genes involved in the biosynthesis of defense-related metabolites may give insight in their regulatory relationships. As we have particular interest in genes and pathways that are involved in biosynthesis of compounds within the herbivory-induced volatile blend that is of importance for attraction of natural enemies of spider mites, we focused on the expression profiles of a gene encoding a lipoxygenase (LOX) which may be involved in green leaf volatile [46,47] and/or JA biosynthesis [48] and two genes encoding terpene synthases (TPS) that synthesize multiple mono- and sesquiterpenes and were found to be induced upon spider-mite feeding [44]. Furthermore, we included a gene encoding an UDP-glycosyltransferase (UGT), *CsUGT,* putatively involved in the glycosylation of secondary metabolites [49] and a gene encoding a sucrose transporter (SUT), *CsSUT,* both of which were selected as representatives of the genes downregulated by spider-mite feeding. In addition to the gene expression patterns detected in response to spider-mite herbivory, expression profiles of these genes after infection with downy mildew [45] and in different organs of unchallenged plants [12] were included to analyze co-expression relationships with the 119 selected TF genes that show differential expression upon spider-mite herbivory. Hierarchical clustering of these genes was analyzed using Genemaths (www.applied-maths.com) based on log_2_-transformed and mean-centered expression data [44]. Distances between genes and clusters and significance of co-expression were calculated based on Pearson correlation and UPGMA (Unweighted Pair Group Method with Arithmetic Mean). We found that the TF genes most closely co-expressed with *LOX* (*p* = 0.05) are a *HOX*, two *bZIPs* and two *MYBs* (Figure 5). Both spider mite-induced *TPS*s closely co-expressed with each other and with a *bHLH*, a *MYB* and an *ERF* (*p =* 0.04). Furthermore, although not significantly, three other *MYB*s, a *NAC*, a *TFIIIC* and a *TCP*, co-expressed with both *TPS* genes more than other TF genes (*p* = 0.08). *CsUGT* and *CsSUT* co-expressed with an *ERF*, a *bHLH,* a *MYB* and a *WRKY* (*p* = 0.05) (Figure 5).

In addition, the strongest induced TF, i.e., *MYC* (*Csa3M002860*) co-expressed with a gene encoding DXS (1-DEOXY-D-XYLULOSE-5-PHOSPHATE SYNTHASE), and eight other genes that encode proteins with unknown functions (Appendix A). All these genes were upregulated during spider-mite feeding, especially in accession Corona. Seven genes co-expressed with the most strongly suppressed TF *bHLH* (*Csa3M002860*), however protein functions of these genes are mostly unknown except for one gene that encodes a putative lyase.

### 2.6. Motifs in Promoters of Defense-Related Genes

The binding motifs in the promoter regions of spider mite-inducible *CsLOX*, *CsTPS9* and spider mite-suppressed *CsUGT* and *CsSUT* were analyzed to obtain more hints for possible regulators. Multiple motifs were found in each of the promoters by aligning the sequences to PlantCARE, the database of plant *cis*-acting regulatory elements [50]. There were 32 non-redundant elements in the promoter of *CsLOX*, 35 in the promoter of *CsTPS9*, and 24 and 30 in the promoters of *CsUGT* and *CsSUT*, respectively (Figure 6). Thirteen elements were present in all four promoters, including motifs HSE, circadian, Box I, G-box, TTC-rich repeats, CGTCA-motif, AAGAA-motif, TATA-box, CAAT-box, and three unnamed motifs (Figure 6). The promoters of the upregulated genes, *CsLOX* and *CsTPS9*, shared an AE-box, an MBS (MYB Binding Sites) and a motif with unknown function, while there was no common motif shared by the promoters of the downregulated *CsSUT* and *CsUGT*. Every one of these promoters also had multiple specific motifs which were not present in the other three.

## 3. Discussion

A central goal in improving the understanding of the response of plants to biotic stresses is to identify genes that are responsive to the stress, determine how they are regulated and what their role is in the plant under the biotic stress of interest. With the development of genomic technologies applicable for more (crop) species, including methods for gene expression profiling, these issues can be addressed on a more global scale. Here, we examined the transcriptional reprogramming of *C. sativus* TF genes in response to *T. urticae* spider-mite herbivory. We identified 119 TF genes that are differentially expressed upon spider-mite herbivory and we compared expression profiles of these TF genes with the expression profiles that were reported upon infection with an oomycete pathogen. 

### 3.1. TF Genes in the Cucumber Genome 

TFs are key regulators of plant gene expression and often connect phytohormonal signaling pathways to biosynthetic pathways. We used the RNA-seq dataset in which two *C. sativus* genotypes were compared for their transcriptional response upon two-spotted spider mite feeding [44] to study the role of TFs in the response of cucumber to spider-mite herbivory. Identification of TF genes among the DEGs induced by spider-mite feeding may help to understand the complexity of the defense regulatory networks. In order to obtain an overall view on TF genes within the cucumber genome, we first identified all 1212 putative TF genes among the 23,248 predicted cucumber genes according to the feature domains of the encoded proteins. This number is lower than the 1575 putative TF genes identified in the iTAK database [8]. However, annotation of the TF genes collected in our study is based on alignments to published genes in Genbank (http://www.ncbi.nlm.nih.gov/genbank/) and therefore considered to be more reliable. Moreover, the number of differentially expressed TF genes belonging to major defense-regulated TF families such as *MYB*, *bHLH*, *AP/ERF*s, *NAC*, *WRKY* are comparable with the identification used in the present study or based on iTAK annotation. TF genes represent 5.2% (6.7% if iTAK database is used) of the protein-encoding genes in the cucumber genome. This is comparable to the 5.7% of TF genes in tomato [10], but less than the 8.9% (based on iTAK) in Arabidopsis and 10.6% in rice (iTAK).

We noticed that some TFs showed accession-specific regulation by spider-mite feeding. A number of TF genes were strongly induced (Figure 2, group II), whereas others were more suppressed (Figure 2, group III) in Corona (Co) compared to Chinese long (Cl). This matches with the results of our previous study that showed that spider mites feed more intensely on Co than on Cl, with the former consequently emitting more volatiles and attracting more predatory mites [44] than Cl.

Comparing our data set of TFs regulated by spider-mite feeding to the TFs induced by downy mildew [45] revealed that the fungus infection regulated more TF genes (Table 1), of which *WRKY* was also among the strongly affected families. Furthermore, at least 46 TFs were differentially regulated both by spider-mite feeding and downy mildew infection [44], reflecting specific responses of cucumber to different biotic stresses.

### 3.2. TF Genes Responsive to Spider-Mite Feeding

Previously, we found that early spider-mite feeding at a relatively low herbivory pressure resulted in about 10% of the genes being differentially expressed [44]. Likewise, slightly more than 10% of the putatively annotated TF genes significantly changed expression upon spider-mite herbivory. Approximately 21% of the genes encoding WKRYs in the cucumber genome were responsive to feeding by spider mites, making the WRKYs the most affected TF family. The relative contribution of responsive genes within the *AP2/ERF* (16%) and *bHLH* (14%) TF families also superseded the average response level. Furthermore, *MYB* (10%) and *ARF* (10%) gene families displayed an average response level and this may imply that *WKRYs*, *AP2/ERFs* and *bHLHs* play more important roles than other TF gene families in the regulation of defense processes to spider-mite feeding in cucumber. *MYBs*, *bHLHs* and *ERFs* were also in the top three regulated TF gene families in tomato upon spider-mite feeding for different periods ranging from 1 to 24 h, resulting in 187 differentially expressed TF genes among the 2132 DEGs [41]. In contrast to the relative contribution of spider mite-regulated *WRKY* TF genes in cucumber (14 out of 119 TF genes), only nine *WRKY*s (out of 187) were regulated in tomato in response to spider-mite feeding. Transcripts of six of the tomato *WRKYs* increased during the spider-mite herbivory and three decreased.

### 3.3. Possible Regulatory Relationships between TFs and Metabolite Biosynthesis Associated Genes

TFs are key regulators of plant defense. However, TF proteins cannot repel herbivores and the defense of the plant relies on morphological restructuring (such as changes in trichome density) or re-configuration of secondary metabolism. The genes involved in secondary metabolite biosynthesis in plants are likely regulated by TFs and the understanding of these regulatory relationships are core questions to understand mechanisms of plant defense. In cucumber, a number of essential genes that are involved in direct and indirect defense against spider mites have been identified including the pathway genes involved in the biosynthesis of cucurbitacin C, which was shown to be regulated by two bHLH TFs in leaves and fruits, respectively [51], and *CsTPSs* involved in the biosynthesis of terpenoids that are of importance for the attraction of natural enemies of spider mites [47,52,53]. In this study we analyzed the co-expression of these defense-related genes using gene expression data of plants exposed to spider-mite herbivory [44], infection with downy mildew [45] as well as of different organs of cucumber that were not exposed to biotic stress [12]. Although there will be a time delay between the abundance of TFs and the expression of their target genes, their co-expression could indicate putative candidate key-regulators. *CsTPS9* and *CsTPS19* were hardly expressed in any of the organs of cucumber but were induced in leaves upon spider-mite feeding. It is possible that TFs including *MYB*, *NAC*, *AP2/ERF* and *bHLH*, which were only highly expressed in spider mite-infested leaves, play a role in the regulation of the expression of these genes. Twenty-one motifs were found in the promoter sequences of the spider mite-induced *CsTPS9* and *CsTPS19* and 16 motifs were found in the promoters of both spider mite-suppressed genes *CsSUT* and *CsUGT*. Intriguingly, 13 motifs were present in all four promoters analyzed, suggesting binding by similar types of TFs. This would imply that both up- and down-regulation could be the result of binding to (different) members of the same TF family, possibly further fine-tuned by the interaction between different TFs and/or other co-regulators. 

In conclusion, we have shown that TFs play an important role in the response of cucumber to herbivory by spider mites. *MYB*, *bHLH*, *AP2/ERF* and *WRKY* TF genes are among the most regulated TFs during the first days of feeding by spider mites. Furthermore, we identified a number of TFs potentially involved in the upregulation of indirect defense-related genes including *CsLOX* and two *CsTPS*s, as well as the TFs that seem to downregulate metabolism-associated genes such as *CsUGT* and *CsSUT*. Further work to confirm the involvement of these TFs in induced indirect and direct defense of cucumber against spider mites is in progress.

## 4. Materials and Methods 

### 4.1. Plants and Mites

Seeds of bitter accession of Cucumber (*C. sativus*), Chinese long 9930 (CL), were kindly provided by the lab of Dr Sanwen Huang, and seeds of non-bitter accession, Corona (CO) were bought from Monsanto. Seeds were germinated in potting soil and cultivated in a greenhouse compartment (16-h photoperiod (22 ± 2 °C), 8-h night period (18 ± 2 °C)). Two-spotted spider mites (*T. urticae*) were provided by Koppert Biological Systems and reared on Lima bean (*Phaseolus vulgaris*) plants for multiple generations before experiments on cucumber. 

### 4.2. Identification of TF Genes in the Cucumber Genome 

The genome sequence and predicted genes of *C. sativus* L. accession 9930 (Chinese Long) were downloaded from the Cucurbit Genomics Database ([11,12]; version 2.0) for DEG analysis and TF identification. To analyze the RNA-seq data for putative TFs, a literature search was conducted and the sequences were aligned to online databases including Genbank non-redundant protein sequences database and InterPro protein signature databases (http://www.ebi.ac.uk/interpro/). Genes encoding proteins with specific domains for different TF families were identified as TF genes. 

### 4.3. Grouping of the DEGs and Identification of TFs Responses to Feeding by Spider Mites or Downy Mildew

Leaves of cucumber (*Cucumis sativus* accession Chinese Long (Cl, with bitter foliage) and Corona (Co, with non-bitter foliage) were infested with two-spotted spider mites (*Tetranychus urticae*) for different time periods. Samples were harvested and dynamic transcriptome changes were checked by analyzing expression of two indicating inducible defense-related genes *LIPOXYGENASE* (Csa2M024440) and (*E,E*)-α-*FARNESENE SYNTHASE* (Csa3M095040) using qRT-PCR as described in detail in reference [44]. Then representative time points and samples were selected and genome-wide gene expression was analyzed using a RNA-seq approach leading to identification of differentially expressed genes (DEGs) as previously described [44]. To create a self-organizing map (SOM), expression values of DEGs were imported into GeneMaths XT (http://www.applied-maths.com) and normalized to the standard deviation of all the measured points of each gene of different accessions and time points. Pearson correlation was used to compare the similarity of the expression patterns of the genes (http://www.applied-maths.com). The number of nodes in the X-dimension and Y-dimension are both set to 2. The effects of genotype and time point since feeding were tested using a general linear model (GLM) (IBM SPSS STATISTICS, version 22). RPKM (Reads Per Kilobase of transcript per Million mapped reads) values of the genes were set as dependent variables, accessions and days post feeding were set as fixed factors. TFs in the collection of DEGs in response to feeding by spider mites were identified as responsive TF genes and were selected for further analysis. For comparison of cucumber TF transcriptional responses to spider-mite feeding with such responses to downy mildew infection, the TF genes in the corresponding collection of DEGs were selected. The list of DEGs responsive to downy mildew and their expression information were obtained from published data [45]. Expression data of cucumber organs and DEGs in tomato leaves infested by spider mites are from published datasets [12,41].

### 4.4. Co-Expression Analysis of TF Genes and Plant Defense-Associated Genes

The RPKM values of the selected genes were imported into Genemaths XT (www.applied-maths.com) and expression levels were centralized and normalized to the standard deviation for each gene over the analyzed time points. Different accessions as well as different infestations were analyzed separately for expression patterns of selected genes. For co-expression of TFs with selected possible target genes, expression values of the genes from different accessions, different infestations and different organs [12] were analyzed together. The expression data in these treatments were log_2_-transformed and mean-centered; similarity of expression of the genes was analyzed using Pearson correlation and the genes were clustered using UPGMA (Unweighted Pair Group Method with Arithmetic Mean), which calculates the distance of genes based on their similarity of expression and places them into clusters. More closely co-expressed genes have a shorter distance in the hierarchical clustering diagram [44].

### 4.5. Identification of Binding Motifs in the Promoter Region

The 2000 bp sequences upstream of the start codon of the selected genes (*CsLOX*, *CsTPS9*, *CsUGT* and *CsSUT*) were extracted from the cucumber genome [11] and analyzed online for *cis*-acting elements using the service of PlantCARE (*Cis*-Acting Regulatory Element, [50]).

## Figures and Tables

**Figure 1 plants-09-01014-f001:**
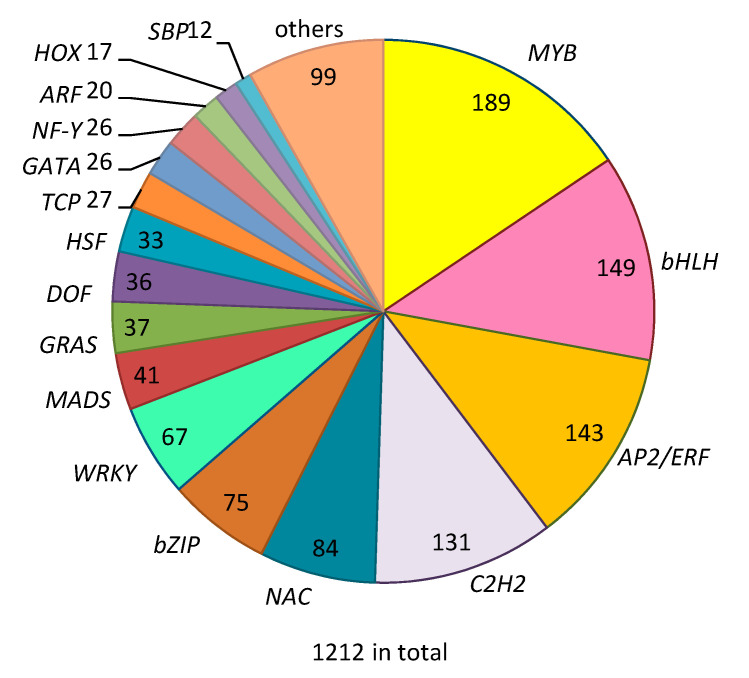
Transcription factor gene families identified in cucumber with numbers indicating number of transcription factor (TF) genes belonging to a particular family.

**Figure 2 plants-09-01014-f002:**
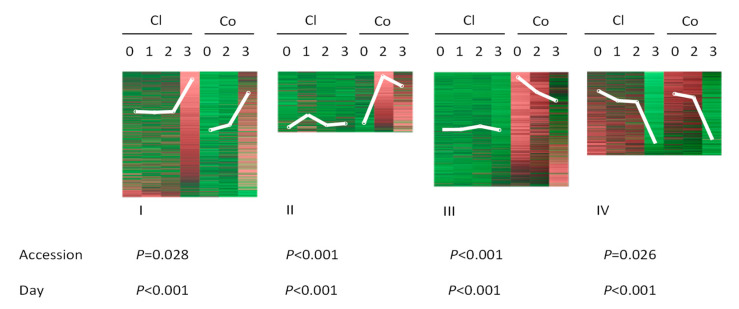
Regulation of gene expression by feeding of spider mites (*Tetranychus urticae*) during one, two or three days in two cucumber accessions (Chinese long (Cl) and Corona (Co), upper part) and effects of accessions and days of feeding on the expression of the differentially expressed genes (DEGs) ((*p*-value), lower part). All DEGs were self-organized using GeneMaths XT (http://www.applied-maths.com) into four groups (I to IV) based on their expression at different time points of feeding. White lines indicate the general expression trend for each group. DEGs from each group were tested separately using a general linear model (GLM) (IBM SPSS STATISTICS, version 22). Upregulated or downregulated abundance of transcripts was set as variable and the accessions and days of feeding by spider mites were set as factors for which the effect was tested.

**Figure 3 plants-09-01014-f003:**
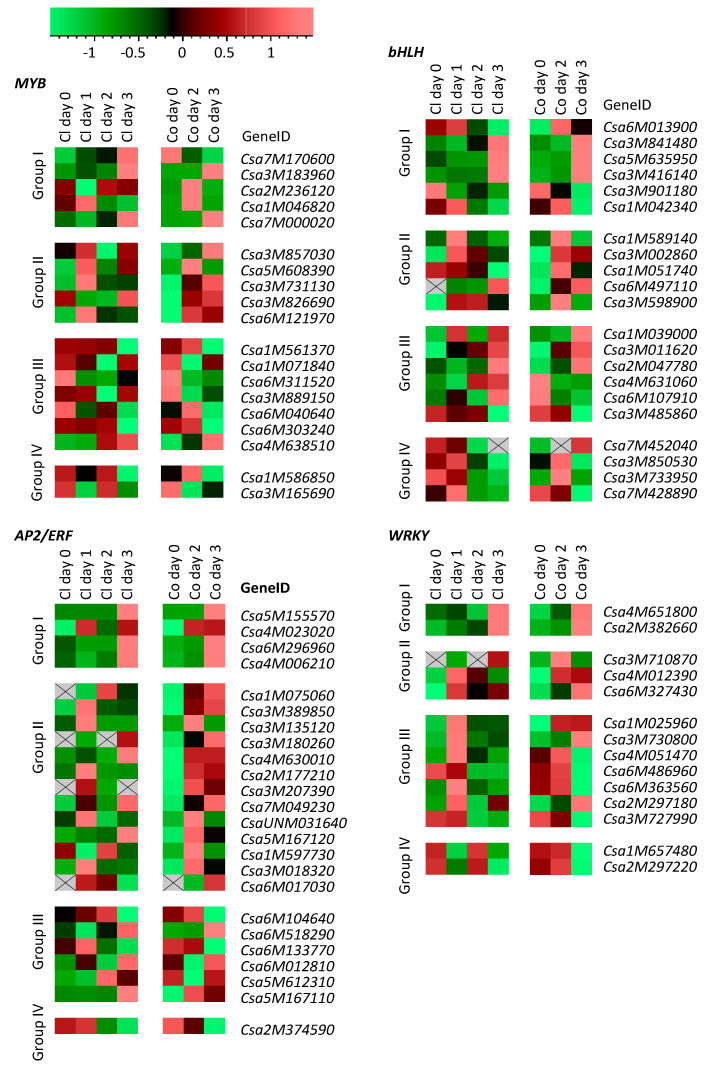
Expression profiling of cucumber transcription factors of four families most differentially regulated by spider-mite feeding (*T. urticae*). Reads Per Kilobase of transcript per Million mapped reads (RPKM) values of the genes obtained from RNA-seq were introduced into Genemaths XT and were normalized to the standard deviation of each gene. Color coding: green represents low expression; red represents high expression. Grey squares with cross indicate that no transcripts were detected.

**Figure 4 plants-09-01014-f004:**
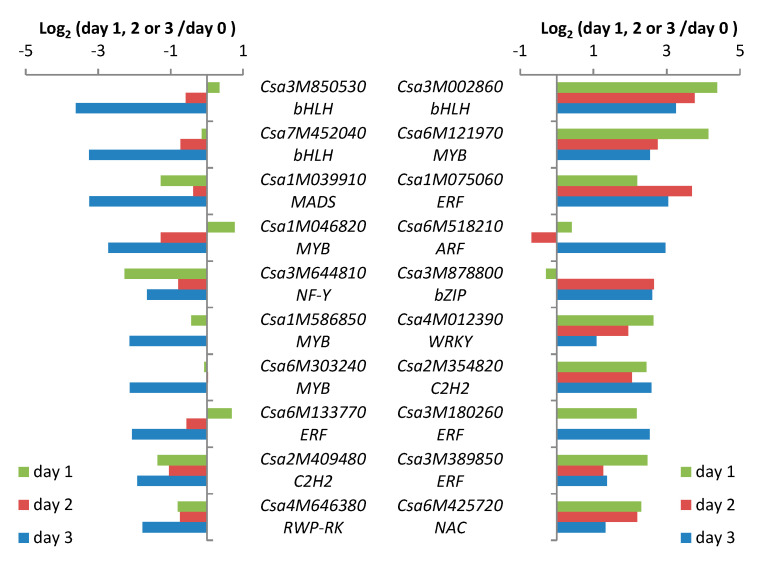
Top 10 strongest down- and up-regulated transcription factor genes in cucumber accession Chinese long after one, two or three days of spider-mite feeding.

**Figure 5 plants-09-01014-f005:**
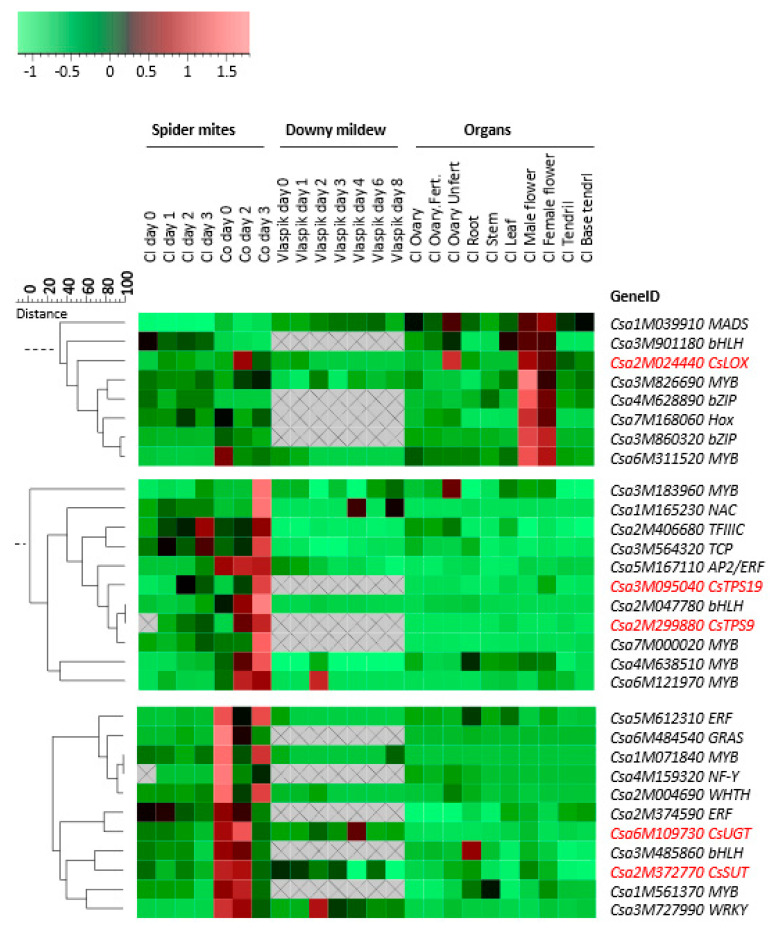
Co-expression of a selection of metabolism-associated genes (indicated in red font, *CsLOX*, *CsTPS9*, *CsTPS19*, *CsUGT*, *CsSUT*) and TF genes during feeding by spider mites (*T. urticae*) [44], downy mildew (*p. cubensis*) [45] infection and in different non-infested/non-infected organs [12] of cucumber accessions Chinese long (Cl), Corona (Co) and Vlaspik. The scale indicates the distance of the similarity of the expression profile of these genes. Color coding: green represents low expression and red represents high expression. Grey areas with cross indicate that no transcripts were detected.

**Figure 6 plants-09-01014-f006:**
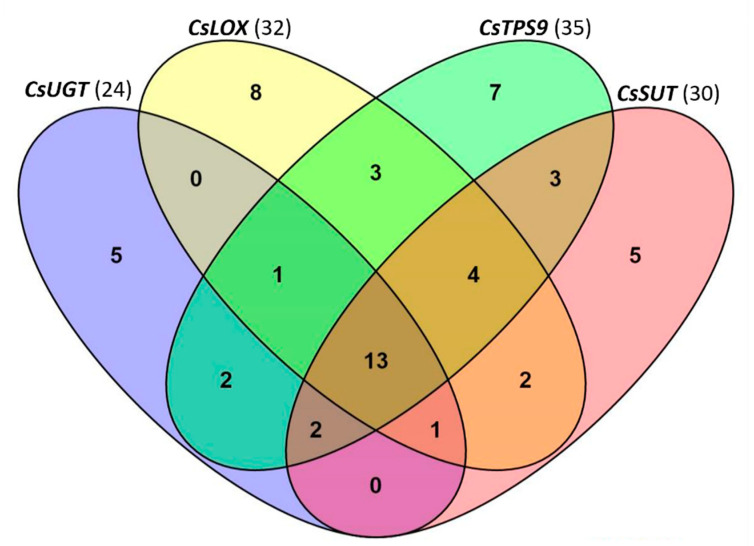
Specific and shared binding motifs found in promoter sequences of *CsLOX* (upregulated), *CsTPS9* (upregulated), *CsUGT* (downregulated) and *CsSUT* (downregulated). Total number of motifs for each gene is indicated in brackets following the gene name.

**Table 1 plants-09-01014-t001:** Number (Nr) of TF genes belonging to different classes/families that are differentially regulated by spider-mite feeding and downy mildew infection or are regulated by both biotic stresses.

Spider Mites							Downy Mildew		Spider Mites and Downy Mildew	
TF Family	Group I	Group II	Group III	Group IV	In Total	% *	Nr	%	Nr	%
*MYB*	5	5	7	2	19	10%	26	14%	5	3%
*bHLH*	6	5	6	4	21	14%	22	15%	6	4%
*AP2/ERF*	4	13	6	1	24	16%	29	20%	14	10%
*C2H2*	1	4	4	1	10	8%	18	14%	2	2%
*NAC*	1	2	3	1	7	8%	28	33%	4	5%
*bZIP*	1	0	2	0	3	4%	15	21%	0	0%
*WRKY*	2	3	7	2	14	21%	31	46%	12	18%
*MADS*	0	0	1	0	1	2%	4	10%	0	0%
*GRAS*	0	0	3	0	3	8%	10	27%	1	3%
*DOF*	0	0	3	0	3	8%	4	11%	0	0%
*HSF*	0	0	1	0	1	3%	7	21%	0	0%
*TCP*	1	0	0	0	1	4%	5	19%	0	0%
*GATA*	1	0	0	0	1	4%	5	19%	0	0%
*NF-Y*	0	0	1	1	2	8%	5	19%	0	0%
*ARF*	1	0	0	0	1	10%	3	15%	0	0%
*HOX*	1	0	1	0	2	12%	0	0%	0	0%
*SBP*	0	0	0	0	0	0%	0	0%	0	0%
others	4	0	1	1	6	6%	18	18%	2	2%
**Total**	**28**	**32**	**46**	**13**	**119**	**10%**	**241**	**19%**	**46**	**4%**

*: Percentage of genes per category that is affected by the indicated stress.

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
