# Peer review of "Genome-Wide Analysis Reveals Transcription Factors Regulated by Spider-Mite Feeding in Cucumber (Cucumis sativus)"

_plants, 2020, doi:10.3390/plants9081014_

Round 1

Reviewer 1 Report

Plants: Manuscript ID plants-879289

Title: Genome-wide analysis reveals transcription factors in response to spider-mite infestation in cucumber (Cucumis sativus)

Authors: Jun He, Harro J. Bouwmeester, Marcel Dicke, and Iris F. Kappers

Brief Summary and Overall Rating:

      The manuscript provides a meaningful contribution to our understanding of the roles of transcription factors in insect infection and the defense mechanism in a crop plant. This manuscript is a continuation of a recently published paper in which the RNA-seq database generated in the previous study is mined and analyzed more thoroughly focusing on the question of those regulatory networks functional in the defense response of cucumber to spider mite infestation.

      The data analyses conducted are appropriate and complete but do require more explanation in one area (see below). The methods are appropriate, however a more complete explanation of the co-expression of selected metabolism associated genes and TFs is needed (see questions below). The paper is highly appropriate for the journal and the data are scientifically and
technically sound, appear repeatable, and mostly appropriately analyzed.

      This paper will be of great interest to pest management specialists who are engaged in improvement of plant defense control mechanisms. The overall merit of the paper is significant and advances our understanding of the behavior of agricultural pests and the plant response systems. The paper is well written and there are few grammatical errors. Accept with minor revisions.

Broad comments

      The introductory material is very complete, informative and provides in depth information on available plant genome sequence data and transcription factors. To my knowledge this work has not been published elsewhere, the methodology is fairly well described with one drawback (addressed in specific comments below). The author’s use of the literature is very good. The methodology is appropriate and understandable. The experimental data is suitable, the statistical analyses acceptable, and the data supports their conclusions.

Specific comments

      The quality of the presentation is very good, however I have a few specific comments that I would like the authors to address before publication.

      A list of abbreviations might be useful for this paper as there are so many.

       Introduction:

      Lines 47-54: This is really the justification and significance most important in this study. A figure of this pathway might be justified but this would be at the discretion of authors-just a suggestion, not required.

       Results:
      Figure 5 and Lines 194-199: It might be due to the extensive amount of data presented in figure 5 but this data is difficult to decipher. In the text, an explanation of how reported Pvalues relate to the figure is needed. The “Hierarchical clustering analysis” need more explanation here and in the methods where it is also not well explained (lines 330-338). The explanation of this method in reference 44 is much better (maybe authors should refer to that paper here). The cladogram has a very broad range of 0-100 and it should be explained how that indicates significant relationships (as opposed to that in Figure S-2 which makes much more sense). It should also be pointed out the figure legend. I understand that the genes in red are of primary significance here but also should be pointed out in the legend. Readers unfamiliar with UPGMA will have difficulty with this data analysis. Authors should provide a better
explanation of why these analyses are used and how the authors interpreted the data.

      Figure 6: What do numbers refer to specifically? The number of motif hits? Please clarify. Is there any reason to point out what the 13 promoter binding motifs present in all 4 genes are? If not, then explain more about the importance the identifying these binding motifs.

       Discussion
      Lines 259-265: I am unsure what the authors are trying to convey in this paragraph. WRKYs are most responsive TF family, then Ap2/ERF and bHLH and then MYB and ARF? The use of the terms “average”, “more than average” and “around or little above average” makes no sense. Why not give the data a numerical property (number, %, or a concrete point)? Could the data be put in a table and then discussed?

      Line 295: Discussion ends abruptly…finish the thought in the last paragraph...why is it important to make this comparison? and what do transcription factors mean in an overall defense to cucumbers and pest management in particular? Tie back to your goals in the introduction with overall significance or next steps or future goals or potential for control
development.

      Marginalia: A few suggested grammatical corrections in the
manuscript where line numbers are highlighted.

Reviewer 2 Report

The manuscript is overall well structured, objective and results are clearly presented, however, discussion could be expanded. For instance, the data presented in Fig. 2 for the groups II and III  could be better discussed in relation to differences between CI and CO. Also, the thoughts regarding gene expression profiles, similarities and differences between mite treated and fungi treated samples will improve the discussion.

 I am also concerned about a few biological replicates used in this study. Generally, it is advised to use  a minimum 3. If it is not possible to add more data to these analyses, at least it should be explained.

It is also unclear the data presented in the third column of Table1  “Spider mites and downy mildew”. What expression data is this? It gives the impression that you have another treatment having both mite and mildew together, however it is not stated in material and methods and not discussed further in results and discussions.  It needs to be clarified in the table heading.
